# Influence of High-Temperature Liquid on Phase Composition and Morphology of Carbothermal Reduction-Nitridation Products from Coal Gasification Slag

**DOI:** 10.3390/ma13061346

**Published:** 2020-03-16

**Authors:** Hudie Yuan, Hongfeng Yin, Yun Tang, Hang Shuai, Yalou Xin, Xilou Pu

**Affiliations:** College of Materials Science & Engineering, Xi’an University of Architecture & Technology, Xi’an 710055, China; yuanhudie@xauat.edu.cn (H.Y.); tangyun@xauat.edu.cn (Y.T.); shuaihang031@163.com (H.S.); xylxauater@163.com (Y.X.); pu17691130772@163.com (X.P.)

**Keywords:** coal gasification slag, high-temperature liquid, carbothermal reduction–nitridation, Ca-α-SiAlON

## Abstract

In this paper, the products using three kinds of coal gasification slags as starting materials were obtained via carbothermal reduction-nitridation at 1450 °C. The effects of high-temperature liquid on the phase composition and morphology of the samples were investigated by XRD and SEM, while the content of high-temperature liquid was calculated by the computer software package FactSage. The results show that: (1) the existence of high-temperature liquid phase is beneficial to the formation and growth of Ca-α-SiAlON phase; (2) The formation of long-columnar Ca-α-SiAlON were greatly affected by the content and viscosity of liquid phase, which is in non-linear relationship with aspect ratios of Ca-α-SiAlON. Among the three kinds of slags, the HT slag with relatively high liquid phase content and the lowest viscosity is the most favorable to the growth of elongated Ca-α-SiAlON grain; the aspect ratio of the formed Ca-α-SiAlON is the largest; Compared to the SH slag with the highest liquid phase content and viscosity, Ca-α-SiAlON prepared from TE slag possesses the smallest aspect ratio, which exhibits equiaxed grain morphology.

## 1. Introduction

Coal gasification slag is an inevitable byproduct from entrained-flow coal gasification [1], which is comprised of coal ash, fluxing agent and residual carbon. According to the literature, the studies on coal gasification slag mainly focused on three aspects: (1) general characterization of the slags, such as composition, morphology, particle size distribution and thermal expansion property [2,3,4]. In addition, in order to improve the efficiency of coal gasification, residual carbon in the slag was investigated extensively [5,6,7]; (2) flow properties and rheological behavior of the slag, which determine the slag tapping temperature in the gasifier [8,9,10,11,12]; (3) the utilization of the slag. Accordingly, the environmentally safe utilization of the waste slags must be addressed and developed. Recently, most of the slags were conducted in cement (39.3%), landfill (33.4%), road and flyover (5.0%), agriculture (2.9%), bricks and tiles manufacture (12.3%) [13]. However, special reports on the SiAlON ceramics prepared by the slags were very scarce, except for that of Yin et al. [14]. Nowadays, it was considered that the comprehensive utilization of the slags is becoming an urgent issue. As we know, the slags can be reused as a secondary resource depending intensively on its unique physico-chemical characteristics. In our previous study on the basic characterization of coal gasification slag, it was found that the slag mainly nicontains SiO_2_, Al_2_O_3_, CaO and C, which could be used to synthesize Ca-α-SiAlON by carbothermal reduction nitridation (CRN) method [15], which is a simple and low-cost method to compound nitride powders [16,17]. More importantly, the nitrogen source for CRN of coal gasification slag could be self-provided by air separation unit in IGCC system. Therefore, it is of great interest to make use of gasification slag for the preparation of SiAlON ceramics.

α-SiAlON has attracted intensive attention because of its high hardness, good corrosion resistance, excellent thermal shock resistance and reduced grain-boundary phases [18]. However, its application is limited due to the low fracture toughness associated with the commonly observed equiaxed grain morphology. Hence, numerous efforts have been made to fabricate new tough α-SiAlON ceramics [19]. It has been found that carefully controlling the nucleation and grain growth facilitates the formation of elongated α-SiAlON grains and consequently improves the fracture toughness of α-SiAlON ceramics. And as a result, it has been widely used in iron and steel metallurgy, ceramics, aerospace and other industries [20,21].

However, synthesizing Ca-α-SiAlON via CRN, with coal gasification slag as raw material, has particularity, which manifested in: (1) Coal gasification slag, one of the solid wastes, is composed of amorphous aluminosilicate glass phases; (2) SiAlON, using natural raw materials or solid wastes as main material, is generally performed by CRN at 1300~1550 °C [22] when the amorphous aluminosilicate phase will be partially or completely converted into a high-temperature liquid phase, which will greatly affect formation, grain growth, and phase transition of SiAlON. At present, there are few works on the influence of high-temperature liquid on the synthesis of Ca-α-SiAlON. Therefore, this paper takes three kinds of gasification slags from different gasifiers, respectively, and the natural mineral pyrophyllite as a reference. Analyzing its chemical composition and the microstructure, the three slags and pyrophyllite were carbothermally reduced and nitrided in high-purity nitrogen at 1450 °C. The nitridation products were characterized to compare the influence of the existence of a high-temperature liquid phase, the amount of liquid phase as well, on its phase and morphology. This experiment not only fills the gaps on the influence of the high-temperature liquid on synthesizing SiAlON powder from the solid wastes but also offers a significance theoretical guidance on the mechanism of the nitridation products from coal gasification slag and obtention of high-content and long-columnar Ca-α-SiAlON powder as well.

## 2. Experimental Part

### 2.1. Experimental Materials

The source of coal gasification slag and pyrophyllite is listed in Table 1. According to the type of gasifier, the slags are named HT, TE, and SH, respectively; and pyrophyllite is also labeled as PRL. The main chemical compositions of slag and pyrophyllite are shown in Table 2. In addition, when comparing the effect of liquid phase on the CRN products, the same mass ratio of CaO: SiO_2_: Al_2_O_3_: C must be ensured; thus, appropriate amount of calcite and carbon black are used to adjust. Meanwhile, calcite was originated from Guangxi Hesen Calcite Co.Ltd. (Chongzuo, China) with a purity of 98.5%.

### 2.2. Experiment Method

#### 2.2.1. Thermodynamics Calculation of Liquid Phase Quantity and High-Temperature Treatment of Raw Materials

In order to determine the quantity of the high-temperature-liquid, Equilib module of Factsage was employed under a standard atmospheric pressure. According to its quality percentage of components, the system total mass was set as 100 g, and the temperature was from 1150 to 1400 °C with 10 °C each interval.

Upon finishing the theoretical simulation, the comparison experiment was carried out. The slags were dried in an oven at 110 °C for 12 h, then placed in a high-temperature furnace and heated in an air atmosphere at temperatures of 1150, 1200, 1250, 1300, 1350, and 1400 °C for 3 h, respectively. The fired samples were quenched, and then the phase compositions were examined.

#### 2.2.2. CRN Process

When comparing the presence or absence of liquid phase, PRL and TE were used as the research object. The raw materials were milled in de-ioned water using Si_3_N_4_ milling media in a planetary ball mill for 6 h; the slurry was dried and sieved through a 180 m sieve. With white dextrin as binder, the powder was pressed into cylindrical sample with a diameter of Ф10 mm, then placed in an alumina crucible and subjected to a carbothermal reduction and nitridation reaction in a horizontal electrical furnace in a high-purity nitrogen atmosphere with a flow rate of 0.4 L/min at 1450 °C for 2 h. Furnace heating rate was about 5 °C/min.

As for the different quantity of liquid phase, HT, TE, SH were used as the research object. Accordingly, to make full use of the residual carbon, Si/C mole ratio was 2:5 for charge calculation. The quantity of the SH carbon was in shortage only, so the suitable amount of carbon should be added. With the same method as the above, the three slags were processed, of which nitriding temperature is 1450 °C and holding time is 4 h with flow rate of N_2_ at 0.4 L/min.

#### 2.2.3. Characterization

Phase identification of the above fired samples and the CRN products was performed via X-ray diffractometry (XRD, using a D/Max 2550V, Akashima, Japan) applying CuK_α1_ (*λ* = 1.5405981 Å) radiation. The morphology of nitridation samples was investigated by scanning electron microscopy (SEM, model Auriga, Zeiss, Oberkochen, Ostalbkreis, Germany). Findit software (2009) combined with Moud software was employed to do the quantitative analysis of mineral phases by fitting methods [23,24].

## 3. Results and Discussion

### 3.1. Influence of the Existence of High-Temperature Liquid Phase on CRN Products

In this paper, the influence of the existence of high-temperature liquid phase on the phase composition of CRN products was studied as follows: As PRL could decompose at about 1000 °C, the products were mullite and quartz, whose reaction equations are shown below.
Al_2_O_3_·4SiO_2_·H_2_O → Al_2_O_3_·4SiO_2_ + H_2_O(1)
3(Al_2_O_3_·4SiO_2_) → 3Al_2_O_3_·2SiO_2_ + 10SiO_2_(2)

It was precisely because there was no liquid phase when CaO and carbon black were added, PRL could be used as a blank sample, Figure 1 shows the phase specimens from PRL and TE when applying the holding time of 2 h with the nitridation temperature 1450 °C. The specimen from PRL presented β-SiAlON and Ca-α-SiAlON phases, whereas, for TE, Ca-α-SiAlON became the dominant phase instead of β-SiAlON, associated with FeSi_x_ and anorthite phases. According to our previous study [25], during the CRN of gasification slag, Ca–Si–Al–O liquid was formed due to the melting of Ca-Si-Al-O glass above the eutectic temperature of 1170 °C (Equation (3)). Then with the increase of temperature, the CaAl_2_Si_2_O_8_ and O-SiAlON phases were obtained at ~1350 °C (Equations (4) and (5)). With the continuous nitridation, O-SiAlON was transformed into nitrogen-rich SiAlONs, such as β-SiAlON and 15R, and finally into Ca-a-SiAlON (Equations (6)–(9)). CaAl_2_Si_2_O_8_ could also be further converted to SiAlON phase in the presence of reducing agent C (Equation (10)). In addition, the appearance of FeSi_x_ was the result of Fe_2_O_3_ contained in the gasification slag. The main reaction process was listed as follows:Ca-Si-Al-O glass → Ca-Si-Al-O liquid(3)
Ca-Si-Al-O liquid + C + N_2_ → O-SiAlON(4)
Ca-Si-Al-O liquid → CaAl_2_Si_2_O_8_(5)
O-SiAlON → β-SiAlON(6)
O-SiAlON → 15R(7)
β-SiAlON → α-SiAlON(8)
15R → α-SiAlON(9)
CaAl_2_Si_2_O_8_ + C + N_2_ → Ca-α-SiAlON(10)

Considering the identical relative content of the main components, it was declared that the TE was even more biased in favor of the generated Ca-α-SiAlON. In order to further determine the content of nitriding products, quantitative analysis was performed using Findit and Moud software. The results were shown in Table 3. The mass percentage of Ca-α-SiAlON phase in the nitridation product of PRL was 61.65 wt. % and β-SiAlON was 38.35 wt. %, while that of Ca-α-SiAlON was 99.99 wt. %, which was obviously more than that of PRL. Besides, in TE, the mass percentage of CaAl_2_Si_2_O_8_ and FeSi_x_ was all 0.002 wt. %, which may be due to the software errors, including the algorithm errors.

In the nitridation products of PRL, the mass percentage of Ca-α-SiAlON and β-SiAlON were 61.65 wt. % and 38.35 wt. %, respectively. In contrast, the mass percentage of Ca-α-SiAlON in the nitridation products of TE was 99.99 wt. %, which showed obvious increase in content compared to that of PRL. Meanwhile, the 0.002 wt. % content of CaAl_2_Si_2_O_8_ and FeSi_x_ may be due to the existence of software errors, such as algorithm errors. All of this shows that the system with liquid phase participation was more conducive to the formation of Ca-α-SiAlON phase as well.

Figure 2 shows the microstructure of nitridation products obtained from the PRL and TE at 1450 °C for 2 h. According to Figure 2a and the EDS analysis, there were small particles Ca-α-SiAlON and filamentous β-SiAlON in the nitridation products of PRL, which was consistent with XRD results. At the same time, it can be seen from Figure 2b that the Ca-α-SiAlON grains are larger, and the particles are long and thick. In general, the morphology of the crystal depends on the growth environment and crystal habit of the crystal itself that is deemed to be the most essential factor. According to Wulff theorem [26,27,28], when it is to be in equilibrium, the crystal will adjust its shape to minimize the total free interface energy of its own. Therefore, the growth habit of the crystal determines that its exposed surface is a surface with a lower interface energy as much as possible. The final morphology of the crystal growth depends on the relative growth rate of each crystal surface, which is governed by the growth driving force that is derived from the supersaturation or undercooling of the growth environment [29,30,31]. However, PRL almost has few liquids at high temperature, which caused difficulty in mass transfer in the environment and largely growth-driving force. In this case, it was easy to form a large amount of α-SiAlON crystal nuclei and make the crystal nuclei to be together in the CRN process, which leads to collide with and inhibit with each other among the nuclei; thus, only the equiaxed grain would be obtained. In contrast, TE could form more liquid at high temperature, in one hand, it made the grain to spread to the interface that was suitable for the growth, on the other hand, the grain growth-driving force was reduced, which could ensure the growth of the grain crystallization according to its own habits, so the long column of α-SiAlON could be easy to obtained. Therefore, the existence of high-temperature liquid phase was not only beneficial to the generation of Ca-α-SiAlON but also was more inclined to grow into a long column Ca-α-SiAlON.

### 3.2. Effect of the Content of High-Temperature Liquid Phase on CRN Products

The amount of liquid phase was calculated by Equilib module of Factsage software, and the results were shown in Figure 3. With the increase of temperature, the content of liquid phase showed an increasing trend. When the temperature increased to 1350 °C, the system tended to be balanced and the content of the liquid phase was almost the most and stayed the same. In addition, it was evident from Figure 3 that the three types of the slags showed significant difference in the amount of liquid phase, which was represented by m(TE) < m(HT) < m(SH), and the content in SH was Up to 90 wt. % or more at 1400 °C. Meanwhile, in terms of the simulated calculation, the liquid phases were mainly a calcium aluminosilicate glass (Ca-Si-Al-O) phase, which was consistent with the literature [25,32].

Because of complexity related to the raw materials (gasification slags) system, the calculated theoretical liquid phase amount and the corresponding temperature may deviate from the experimental results. Therefore, the raw materials were ground to a fine powder of less than 180 meshes and pressed into a Φ 10 × 10 mm cylindrical sample. Then, the samples were placed in a high-temperature furnace at varied temperatures from 1150 to 1400 °C for 3 h and quenched.

The XRD patterns of the quenched samples from TE, HT and SH that were heated at 1150–1400 °C for 3 h were shown in Figure 4. It was observed that the phases of samples were dominated by anorthite (CaAl_2_Si_2_O_8_) and a small amount of quartz (SiO_2_) at 1250 °C. With the increase of temperature, the content of CaAl_2_Si_2_O_8_ and SiO_2_ gradually decreased. When the temperature was increased to 1300 °C, the crystal phases in the TE disappeared, which indicated that TE completely melted into the liquid phase at 1300 °C. In contrast, the completely melting temperatures of HT and SH were relatively higher, around 1350 and 1400 °C, respectively.

In order to further study on the effect of the amount of liquid phase on the CRN products, the above three kinds of slags were selected and heated in high-purity nitrogen with a flow rate of 0.4 L/min at 1450 °C for 4 h. The phases of the products were analyzed, and the XRD pattern was shown in Figure 5. It was shown that the main crystalline phases of the three products were Ca-α-SiAlON, β-SiAlON; 15R and CaAl_2_Si_2_O_8_ were the minor phase, and all samples presented a small amount of FeSi_x_ phase.

The results of quantitative analysis by Findit software and Moud software were shown in Table 4. It could be seen that the content of Ca-α-SiAlON from TE was the most, of which the mass percentage was up to 98 wt. %, and that from HT was the lowest, which indicated that the more liquid phase, the more favorable to Ca-α-SiAlON content.

Figure 6 showed the microstructure of the nitridation products obtained from the three slags in high-purity nitrogen with a flow rate of 0.4 L/min at 1450 °C for 4 h. As shown in Figure 6a–c, it could be seen that Ca-α-SiAlON in the nitridation products from TE with the lowest content of high-temperature liquid phase was equiaxed, and the grains were numerous and dense. For HT, with more content of the liquid phase than TE, the Ca-α-SiAlON grains were long columnar, and the aspect ratio of them was significantly increased. However, for the SH slag with the most liquid phase, the Ca-α-SiAlON grains in the products did not continue to increase in the length. Instead, it was increased in radial only, and the aspect ratio decreased. That was because the crystal growth habit of α-SiAlON was anisotropic, and there was no direct relationship between the number of crystal nucleus and formation long grains of α-SiAlON. Whether it could grow to be the columnar grain depended on its dynamics in the process of grain growth conditions that was the suitable growth driving force, which depended on two factors. One was the diffusivity of the liquid phase, and the other was the supersaturation of the liquid phase at the interface of solid–liquid reaction. In the early stage of the research team, the high-temperature viscosity of slag was studied [33]. It was found that the viscosity of HT slag was the lowest, and the SH was the largest. In other words, in the liquid phase for HT slag, both a large amount and a low viscosity, the crystal grains possessed great ability to diffuse, and the driving force for the growth of the crystal grains for HT slag was lower. That might be why the α-SiAlON grains were likely to grow into long grains, which were expressed as large length–diameter ratio, whereas the content of high-temperature liquid for TE was not enough, which resulted in the mass transfer being difficult and the driving force for the growth increasing. According to Jackson interface theory [34,35], when the growth-driving force of the crystal in the growth system is too large, the interface might become rough. In that case, if the growth habit of crystals turns isotropic from anisotropic, the Ca-α-SiAlON grain will grow to be equiaxial grain. For the SH slag, the amount of liquid phase was the most, but the viscosity was the largest; the particle diffusion ability was reduced, and the driving force for growth of the crystal was increased. However, the XRD pattern of nitridation product from SH in Figure 5 shows that the number of the kinds of the crystal phase was the highest; it was similar to the introduction of other seed crystals in the Ca-α-SiAlON grain, which could reduce the growth-driving force of Ca-α-SiAlON crystal grains. Consequently, length-diameter ratio of Ca-α-SiAlON from SH was smaller than that from HT, but it tended to increase compared with TE.

## 4. Conclusions

(1)The presence of a high-temperature liquid phase not only favored the formation of the Ca-α-SiAlON phase but also produced a Ca-α-SiAlON that tended to grow into a columnar shape.(2)The relationship between the amount of liquid phase and the aspect ratio of Ca-α-SiAlON phase was nonlinear. The morphology of crystal grains was not only related to the amount of liquid phase but also might be affected by the viscosity of the liquid phase.(3)The HT with many liquid phases (~65 wt. %) and low viscosity (~1.99 Pa·S) was most conducive to the growth of elongated Ca-α-SiAlON; and for the SH with the highest content of liquid-phase (~95 wt. %) and the maximum viscosity (~3.37 Pa·S), the Ca-α-SiAlON showed the largest aspect ratio; while Ca-α-SiAlON grains fabricated by TE slag with the lowest content of the high-temperature liquid (~50 wt. %) and the medium viscosity (~2.81 Pa·S) were of equiaxed shape.

## Figures and Tables

**Figure 1 materials-13-01346-f001:**
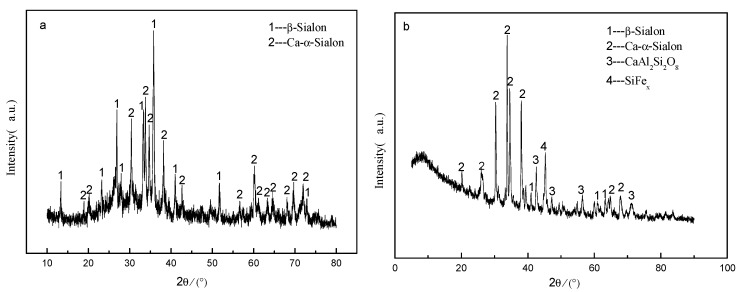
XRD images of the samples prepared from PRL(**a**) and TE (**b**) at 1450 °C for 2 h.

**Figure 2 materials-13-01346-f002:**
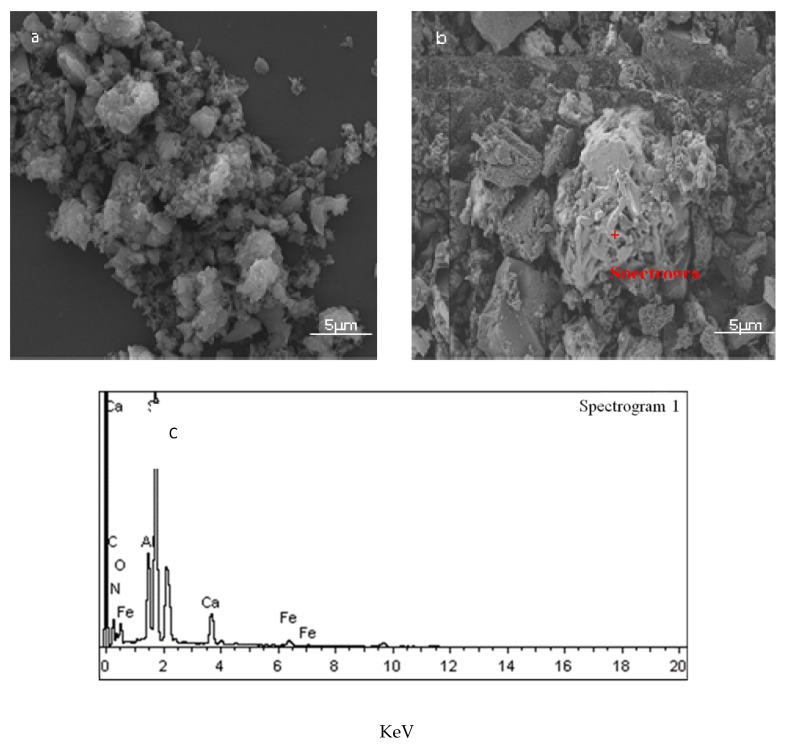
SEM and EDS (**c**) images of the samples prepared from PRL (**a**) and TE (**b**) at 1450 °C for 2 h.

**Figure 3 materials-13-01346-f003:**
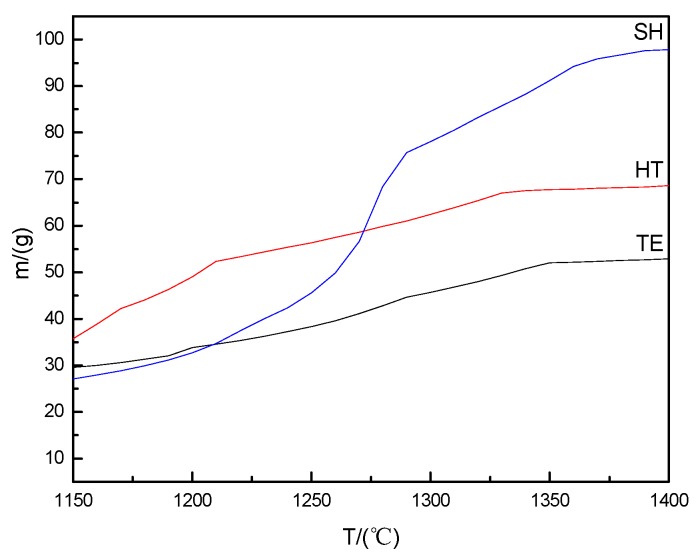
The trend of high-temperature liquid phase with the change of temperature.

**Figure 4 materials-13-01346-f004:**
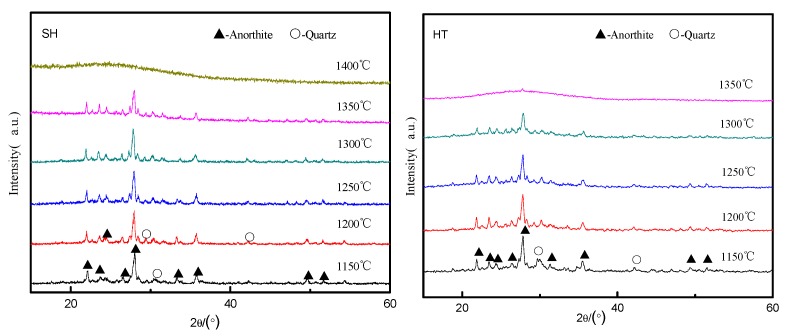
The phase composition of the slags at high temperatures.

**Figure 5 materials-13-01346-f005:**
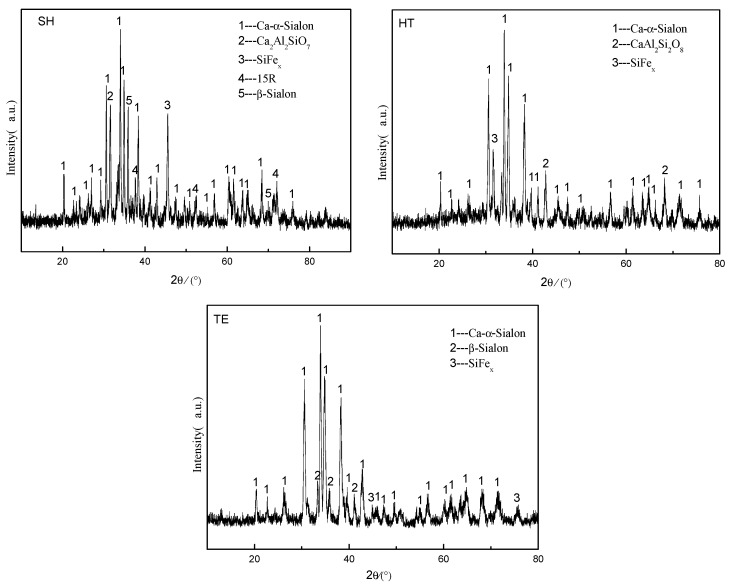
XRD images of the products sintering at 1450 °C for 4 h.

**Figure 6 materials-13-01346-f006:**
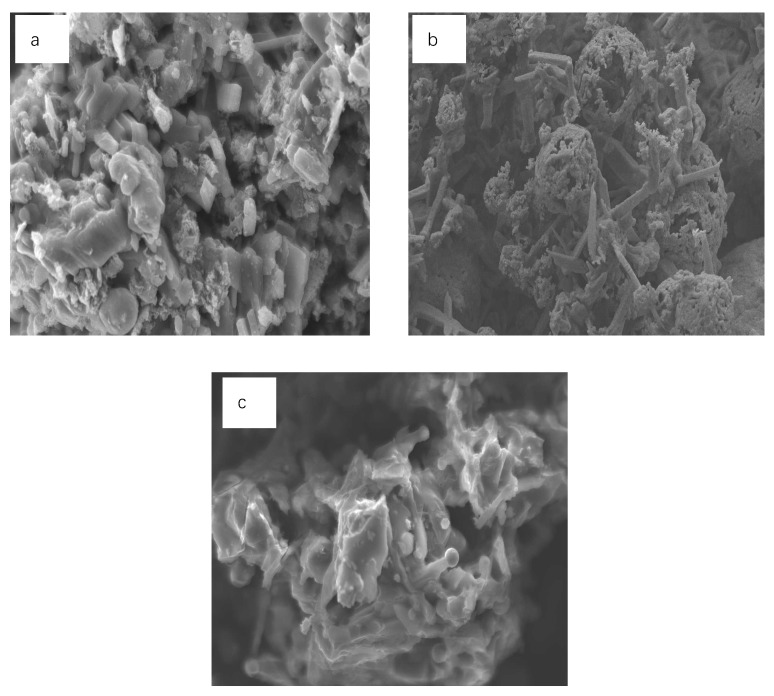
SEM images of the nitridation products prepared from TE (**a**), HT (**b**) and SH (**c**) at 1450 °C for 4 h.

**Table 1 materials-13-01346-t001:** Source of raw materials.

Material	Gasifier	Trade Names
HT	HT-L	Luxi Chemical Group Co. Ltd. (Liaocheng, Shandong, China)
TE	Texaco	Shaanxi Shenmu Chemical Industry Co.Ltd. (Yulin, Guangxi, China)
SH	SHELL	Yiyi Coal Group Kaixiang Chemical Co. Ltd. (Sanmenxia, Henan, China)
PRL	-	Zhejiang Qingtian Ye Wax Stone Co.Ltd. (Lishui, Zhejiang, China)

**Table 2 materials-13-01346-t002:** Main chemical composition of raw materials.

Material	Chemical Composition (wt. %)								
	CaO	SiO_2_	Al_2_O_3_	Fe_2_O_3_	Na_2_O	K_2_O	MgO	TiO_2_	C
TE	5.53	31.50	11.58	3.12	0.77	1.83	0.94	0.42	37.25
HT	15.04	32.82	12.25	5.41	0.66	0.96	0.9	0.44	27.99
SH	10.35	51.53	25.05	7.49	0.58	1.03	2.51	0.88	0.71
PRL	-	67.50	21.90	\0.50	-	0.40	-	0.30	-

**Table 3 materials-13-01346-t003:** The result of quantitative phase analysis.

Sample	Quality Percentage/(wt. %)			
	Ca-α-SiAlON	β-SiAlON	CaAl_2_Si_2_O_8_	FeSi_x_
PRL	61.65	38.35	-	-
TE	99.99	-	0.002	0.002

**Table 4 materials-13-01346-t004:** The result of quantitative phase analysis at 1450 °C for 4 h.

Sample	Quality Percentage/(wt. %)					
	Ca-α-SiAlON	β-SiAlON	CaAl_2_Si_2_O_8_	Ca_2_Al_2_SiO_7_	15R	FeSi_x_
TE	97.64	2.34	-	-	-	0.02
HT	66.48	-	33.07	-	-	0.45
SH	78.64	15.43	-	-	0.70	4.03

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
