# Peer review of "Influence of High-Temperature Liquid on Phase Composition and Morphology of Carbothermal Reduction-Nitridation Products from Coal Gasification Slag"

_materials, 2020, doi:10.3390/ma13061346_

Round 1

Reviewer 1 Report

The aim of this paper to assess the influence of high-temperature liquid on the synthesis of Ca-α-SiAlON, starting from three kind of gasification slags from different gasifiers.

As structures, some minor corrections are required,a sin the following :

1) Introduction: Could be slightly extended. Furthermore, especially on pag 2 some references can be applied on the sentences.

2) Results and discussion. Figure 2. XRD pattern need to be better explained, assessing all phases appeared and qualitatively discuss their ratio. EDS spectra of the elongated and spherical particles must be also added

3) Based on the discussed results, the reaction process need to be proposed, extending the comparison of the three gasification slugs based on it.

4) As possible, the conclusion, should have quantitative results.

Reviewer 2 Report

This is a nice work on understanding the effect of phase composition during the CRN process on the characteristics of the products is presented. The work appears technically sound, however it can be further improved if the following comments are addressed:

1)The authors have not provided any significant information about thermodynamics of the process. Some basic details should be provided.

2) They have mentioned several software packages such as Factsage, Findit and Mood but have not referenced them properly. Appropriate references should be added.

3) What is "rom" in section 2.2.1? This probably a type and should be corrected.

4) "X" should be subscript in FeSix in the text.
